# CriTER-A: A Novel Temperature-Dependent Noncoding RNA Switch in the Telomeric Transcriptome of *Chironomus riparius*

**DOI:** 10.3390/ijms221910310

**Published:** 2021-09-24

**Authors:** Cristina Romero-López, Alfredo Berzal-Herranz, José Luis Martínez-Guitarte, Mercedes de la Fuente

**Affiliations:** 1Instituto de Parasitología y Biomedicina López-Neyra (IPBLN-CSIC), Av. Conocimiento 17, 18016 Armilla, Granada, Spain; aberzalh@ipb.csic.es; 2Grupo de Biología y Toxicología Ambiental, Facultad de Ciencias, Universidad Nacional de Educación a Distancia (UNED), 28230 Las Rozas, Madrid, Spain; jlmartinez@ccia.uned.es; 3Departamento de Ciencias y Técnicas Fisicoquímicas, Universidad Nacional de Educación a Distancia (UNED), 28230 Las Rozas, Madrid, Spain

**Keywords:** telomeric noncoding RNAs, RNA thermo-switch, RNA structure prediction, conformational analysis, SHAPE analysis

## Abstract

The telomeric transcriptome of *Chironomus riparius* has been involved in thermal stress response. One of the telomeric transcripts, the so-called CriTER-A variant, is highly overexpressed upon heat shock. On the other hand, its homologous variant CriTER-B, which is the most frequently encoded noncoding RNA in the telomeres of *C. riparius*, is only slightly affected by thermal stress. Interestingly, both transcripts show high sequence homology, but less is known about their folding and how this could influence their differential behaviour. Our study suggests that CriTER-A folds as two different conformers, whose relative proportion is influenced by temperature conditions. Meanwhile, the CriTER-B variant shows only one dominant conformer. Thus, a temperature-dependent conformational equilibrium can be established for CriTER-A, suggesting a putative functional role of the telomeric transcriptome in relation to thermal stress that could rely on the structure–function relationship of the CriTER-A transcripts.

## 1. Introduction

Telomeres are complex and dynamic structures consisting of DNA repeats, proteins, and telomeric transcripts. The transcriptional activity of telomeres received great attention after the discovery of the telomeric repeat-containing RNA (TERRA) molecules in mammals. They are regulated nuclear noncoding RNAs (ncRNAs) transcribed from telomeres that overturned the longstanding belief that telomeres were transcriptionally silent [1,2]. These telomeric RNA molecules have been reported in several eukaryotes, including metazoans and nonmetazoans, and their essential roles in genome integrity, chromatin regulation, telomere maintenance, and regulation of nontelomeric gene expression have been widely recognised [3,4,5,6,7,8,9,10,11]. Furthermore, an interplay between telomeric RNA expression and the preservation of telomere integrity during cellular stress has been suggested [10]. Notwithstanding the considerable interest of all these issues, the underlying molecular mechanisms are not well understood.

Although all telomeres perform similar functions, telomeric organisation is diverse. Besides the widely conserved telomeric model consisting of tandem arrays of short G/C-rich repeats maintained by specific enzymes, named telomerases, there are alternative telomere structures with different types of telomeric DNA. This is the case with telomeres of dipterans. One of the best known, *Drosophila melanogaster*, shows transposons in the telomeres [12,13]. Another example is the case of nonbiting midges. Several representatives of this group have been described to contain repetitive sequences, between 176 and 350 bp in length, in their telomeres [14,15,16,17,18]. All these findings support the variability in telomere structure and organisation.

*Chironomus riparius*, also known as *Chironomus thummi*, possesses one of these noncanonical telomeres consisting of 176-bp-long tandemly repeats at their chromosome ends. Early analysis of its telomeres showed the existence of transcriptional activity that could be detected after thermal stress as a giant telomeric puff remarkably observed in the right telomere of chromosome 3 (hereinafter ‘III-R’). Activation of transcription under heat shock was also detected in other telomeres but to a lesser extent [19]. Later, it was shown that other stress conditions, such as hypoxia and chemical stress, can also induce the transcriptional activity of these telomeres [20,21,22]. The isolation and identification of repeated sequences allowed for examination of the putative molecular basis of transcriptional activity [14]. The sequences are not identical in all the telomeres though they show a high conservation rate, near 90% of sequence conservation [23]. Three different subfamilies of telomeric sequences, named TsA, TsB, and TsC, were described, showing a specific location in the telomeres [23]. TsA is present only in the III-R telomere, while TsC is only present in the IV-R telomere. TsB is the most widespread, being shared by the remaining telomeres except the telocentric end of chromosome 4 (IV-L) (Figure 1). According to the above, it was recognised that TsA is highly overexpressed under thermal stress conditions whereas TsB also showed transcription but with lower activity. No transcription has been detected for TsC sequences.

Additionally, three different ncRNAs were identified in *C. riparius* that are transcribed from these telomeric sequences: the CriTER-A (*Chironomus riparius* TElomeric RNA from TsA), the CriTER-B (*Chironomus riparius* TElomeric RNA from TsB), and the antisense transcripts produced from a complementary strand of TsB, the so-called αCriTER-B [24]. Further analyses showed that, under environmentally stressful conditions, such as heat shock, levels of CriTER-A increased up to four-fold whereas CriTER-B and αCriTER-B levels rose slightly. Moreover, the level of CriTER-A is much lower when compared with those of CriTER-B and αCriTER-B under normal growing conditions [24]. Experimental and in silico studies revealed the existence of not only common but also specific transcription factor binding sites in TsA and TsB [24,25]. Overall, these findings support the idea of the regulation of transcriptional activity by internal promoters within the repeats besides offering a possible explanation for the differences in transcription observed and for the strong correlation between thermal stress and the relative abundance of different telomeric transcripts.

Revealing the structure and function of telomeric RNA transcripts is crucial for understanding telomere biology and the fundamental biological processes in which they are involved, such as stress response and senescence [10,26]. As far as we know, *C. riparius* is the sole biological system in which two homologous telomeric ncRNAs with substantial and characterized differences in the sequence, CriTER-A and CriTER-B, has been reported, exhibiting each differential transcriptional activity under different environmental conditions. Despite the sequence variation, secondary structure elements are highly conserved in evolution for most functional RNAs. Thus, the potential structural differences between CriTER-A and CriTER-B could result in different functionalities according to environmental conditions, such as thermal stress. To investigate the putative differential structure and function of both these transcripts, a comparative structural analysis was carried out. We hypothesised that despite the sequence homology between these two telomeric transcripts, they exhibit conformational variations that could rely on their different behaviour and relative abundance upon heat shock conditions. Thus, this manuscript provides a structural analysis of the transcripts corresponding to the monomeric units (176 bp), CriTER-A and its homologue CriTER-B. We obtained comprehensive results proving the existence of distinctive folding and conformational behaviour for these two ncRNAs and demonstrating the existence of a conformational switch in CriTER-A, which is associated with temperature changes. These findings support the functional role of CriTER-A in relation to thermal stress and offer novel evidence of the existence of thermosensing structural elements in a stress-inducible telomeric RNA.

## 2. Results and Discussion

### 2.1. RNA Structure Predictions: Remarkably Different Structures for the Two Telomeric Transcripts CriTER-A and CriTER-B

CriTER-A and CriTER-B show high sequence homology, with only ~10% of the variation located in two well-defined regions (Figure 2). It is tempting to propose that the variability in these regions could correlate with certain structural heterogeneity between CriTER-A and CriTER-B transcripts. Furthermore, the transcriptional response to heat shock is not homogeneous in every telomere, rendering a differential abundance of both telomeric transcripts under different temperature conditions, and, presumably, both telomeric transcripts likely undergo structural alterations prompted by temperature change, which may have important functional implications. To test this hypothesis, preliminary theoretical structural models of CriTER-A and CriTER-B monomers were obtained using in silico tools at two different temperature conditions: 20 °C, which resembles the standard physiological situation for *C. riparius* cells; and 35 °C, which entails a thermal stress situation in this biological model (Figure 3 and Appendix A).

In silico models rendered different structural patterns for CriTER-A and CriTER-B. A rod-like structure was predicted for CriTER-A, while CriTER-B was predicted to fold as a Y-shape, including a putative pseudoknot that could compact the overall structure (Figure 3 and Appendix A). The calculations predict a slightly higher thermodynamic stability for CriTER-B than for CriTER-A (Appendix A). Remarkably, CriTER-B folding comprises three stem-loops, two of them highly prevalent in the suboptimal space and involving high base pairing probabilities (Figure 3c and Appendix A). Meanwhile, only one stable, double-stranded region may be identified in CriTER-A (Figure 3a,b). This region emerges as a central, stable core flanked by structural elements that seem prone to conformational changes (Appendix A). In fact, minimum free energy (MFE) structures of CriTER-A calculated at 20 and 35 °C fold into two different hairpin loop structures in this region, whereas changes could not be observed in MFE structures predicted for CriTER-B at both temperatures (Figure 3). Temperature changes on theoretical calculations must be used with caution, since the energy parameters used were measured at 37 °C, and predicting free energy changes other than 37 °C may be prone to significant errors, especially for the highest temperature changes, such as our calculations at 20 °C. Nevertheless, these calculations may be appropriated for comparative purposes, working with two homologues sequences. Thus, these results clearly suggest that both homologous telomeric transcripts, CriTER-A and CriTER-B, fold into a different secondary structure and seem to theoretically display dissimilar temperature-dependent behaviour.

### 2.2. Telomeric RNAs of Chironomus riparius Adopt Two Alternate Conformations

To analyse the conformational heterogeneity of both transcripts, electrophoretic mobility shift assays (EMSA) were carried out. For this purpose, RNA transcripts encoding the natural monomeric CriTER-A and CriTER-B RNA molecules were ^32^P-internally labelled and incubated at 20 (basal temperature for *Chironomus*) and 35 °C (experimental heat shock temperature). Interestingly, different conformers could be detected at 20 °C for both telomeric transcripts CriTER-A (A1, A2) and CriTER-B (B1, B2), with the relative proportion between them remaining constant even at high RNA concentrations (Figure 4). The predominant conformer represents around 80% of the total RNA for both CriTER-A (A1) and CriTER-B (B1), and this relative abundance does not depend on the presence of Mg^2+^ (Figure 4). The existence of molecular species showing different mobility suggests the presence of conformational isoforms: the fast migrating isoforms, corresponding to tightly packed RNA molecules; and the slow migrating isoforms, most likely related to a relaxed, but not unfolded, conformational state. It must be noted that the so-called isoforms B2 are defined by three minority conformers with different electrophoretic mobilities (Figure 4b), showing a constant proportion between them in a concentration-independent manner. However, the A2 conformer behaves as a single molecular specie (Figure 4a).

Importantly, increasing the temperature up to 35 °C promoted a significant increase in the proportion of the A2 conformation of ~4-fold (Figure 4) in a Mg^2+^-dependent way (Figure 4). No differences in the B1:B2 ratio were detected at 35 °C either in the absence or presence of Mg^2+^ (Figure 4b). However, the structural heterogeneity of B2 isoforms was reduced in the absence of divalent cations, suggesting the strong dependence on Mg^2+^ for the acquisition of suboptimal conformations in the CriTER-B variant (Figure 4b). Together, these results suggest that CriTER-A could work as a thermometer by changing its conformation in response to thermal stress conditions.

The existence of slow migrating isoforms could be the result of different structural conformations, but it also could be due to the establishment of intermolecular interactions, leading to the formation of homodimeric complexes in a concentration-dependent manner. In order to test this possibility, ^32^P-internally labelled CriTER-A and CriTER-B transcripts were subjected to competition assays with the respective unlabelled transcripts. These assays were performed in the absence of divalent cations at 20 °C (Figure 5). Importantly, the increase in the concentration of the competitor did not interfere with the formation of the A2 and B2 isoforms, respectively. Instead, high concentrations of the unlabelled CriTER-A transcript induced an evident improvement in the formation of the A2 isoform (Figure 5), in good concordance with previous results (Figure 4a) and confirming that high concentrations of the CriTER-A RNA promote a displacement in the structural equilibrium toward the A2 isoform. No significant changes could be detected in the ratio of the B2 isoform for the CriTER-B transcript (Figure 5), pointing again to the structural homogeneity previously noted for this variant (Figure 4b). All these data support the existence of different structural variants for transcripts CriTER-A and CriTER-B, ruling out the occurrence of intermolecular homodimers.

### 2.3. CriTER-A and CriTER-B Fold into Different Secondary Structures

Previous assays revealed the existence of different conformations for CriTER-A and CriTER-B. Further, the relative abundance of each conformation was influenced by temperature and magnesium conditions for the CriTER-A variant (Figure 4). To map the secondary and tertiary structures of CriTER-A and CriTER-B variants, RNA probing assays were performed at 35 °C using selective 2′-hydroxyl acylation analysed by primer extension (SHAPE) chemistry. This strategy monitors the flexibility of the ribose-phosphate backbone by inducing the formation of covalent 2′-O-adducts in the 2′-OH group mediated by the so-called SHAPE reagents [27].

In a first attempt, we evaluated CriTER-B folding by SHAPE due to its apparent conformational homogeneity with respect to CriTER-A as detected previously (Figure 4). For this purpose, the RNA transcript resembling the monomeric unit of CriTER-B was subjected to NMIA (N-methylisatoic anhydride) treatment, a SHAPE reagent informing about residues that stabilise the architecture of the RNA molecule at small timescales (slow electronic dynamics) [28]. A preliminary overview of the relative reactivity profile showed low values along the entire transcript, suggesting compact folding based on the formation of stable stem-loops with one-sided stacked nucleotides (Figure 6). This observation prompted us to perform further CriTER-B SHAPE analyses with the chemical probe 1M6 (1-methyl-6-nitroisatoic anhydride), which can detect such stacking regions with fast electronic dynamics. The resulting relative reactivity data were compiled with those previously obtained for the NMIA treatment, rendering the complete fingerprinting map of the CriTER-B monomers. This map includes the noncanonical and stacking interactions defining the three-dimensional architecture of the target RNA. This methodology is called SHAPE-dif (differential SHAPE [28]) and has been successfully used for deep structural analysis [28,29].

The application of this strategy confirmed previous predictions. It was outlined that CriTER-B shows high conformational homogeneity, with a compact and likely static structure that results in low reactive positions except for those specific residues placed at the 3′ end of the transcript (Figure 6). Residues exhibiting fast electronic dynamics emerged as the main tuning partners of the CriTER-B conformation (Figure 6). These experimental data were used to predict the theoretical secondary structure model with the ShapeKnots tool [30]. The model reflected a well-defined architecture with three major stem-loops (1 to 3; Figure 6) separated by single-stranded regions. It is noteworthy that stem-loops 1 and 2 are interrupted by internal loops, which are enriched in GA pairs. It is well-known that GA pairs exhibit slow electronic dynamics [28], which is consistent with the observed reactivity pattern. From a functional point of view, GA pairs are usually enriched in functionally relevant regions of RNA molecules. Stem-loop 3 seems to be a continuous stem that is interrupted only by a single-nucleotide bulge (Figure 6). The AU-rich composition of stem-loop 3 supports a certain degree of nucleotide stacking, which is consistent with the SHAPE-dif profile, and might offer a propitious environment for limited structural breathing. This could be reflected by the presence of alternative conformers of CriTER-B, as detected from EMSA data analysis (Figure 4).

Subsequent structural analysis of CriTER-A by SHAPE was independently performed for the two major conformers, A1 and A2, detected at 35 °C. For this purpose, following NMIA treatment, RNA conformers were fractionated by nondenaturing polyacrylamide gel electrophoresis, and the modified residues of each one were detected by primer extension. The observed reactivity patterns revealed the existence of defined highly reactive regions, which significantly differ between the A1 and A2 conformers (Figure 7). These experimental data were used to generate a theoretical secondary structure model with the ShapeKnots tool [30].

The model reflects the existence of two structural conformers, which share a double-stranded core flanked by differentially branched regions, the so-called domains 1 and 2 (Figure 7). Both domains 1 and 2 contain specific subdomains, 1.3 and 2.1, which are also structurally preserved in the two conformations (Figure 7). Importantly, the folding of the core region depends, at least in part, on those variable sequence stretches detected in previous alignment studies (Figure 2), thus pointing to these variations as critical elements for the acquisition of the differential conformation detected for CriTER-A and CriTER-B (Figure 2, Figure 6d and Figure 7d).

In conformation A1, domain 1 presents compact folding with a highly reactive flexible region organised in up to four different subdomains (Figure 7), while domain 2 is a more dynamic region composed of two subdomains folded as short stem-loops organised around a three-way junction (Figure 7). A key feature of isoform A1 is the presence of a double pseudoknot (DPK), which is predicted to involve (i) subdomains 1.3 and 2.1 for pseudoknot PK1 and (ii) subdomains 1.4 and 2.2 for pseudoknot PK2 (Figure 7). Double pseudoknots are compact structural elements, which may operate as protein-recruiting platforms, being important managers in regulatory processes mediated by RNA molecules. We hypothesise that this DPK proposed for the A1 isoform could be related to the functional role of CriTER-A.

In conformation A2, domain 1 modifies its folding towards a more flexible structure, as deduced from the reactivity profile (Figure 7), to generate the so-called domain 1′. It folds into four well-defined stem-loops that determine three different subdomains to those predicted for the A1 isoform, named 1.1′, 1.2′, and 1.4′, and the preserved subdomain 1.3 (Figure 7). From a structural point of view, domain 2′ roughly resembles the predicted domain 2 for the A1 isoform, composed of two stem-loops showing a certain degree of flexibility, as deduced from the reactivity pattern. Domain 2′ also conserves the structural organisation for subdomain 2.1 with respect to isoform A1 (Figure 7). However, a slight sequence slippage induces the extension of the subdomain 2.2′, exposing different nucleotides at the apical loop. The main consequence of this structural tuning is the abolishment of PK2, maintaining the single pseudoknot, PK1, which connects subdomains 2.1 and 1.3 around the core region (Figure 7). This observation makes it appealing to propose that PK2 could be related to a putative thermometer-associated function of CriTER-A, while PK1 could be essential for the maintenance of the structural equilibrium between both isoforms A1 and A2.

## 3. Conclusions

With reference to the hypothesis proposed at the beginning of this study, it is now possible to state that despite the significant sequence homology, CriTER-A and CriTER-B display significant structural differences. Two regions in telomeric sequences accumulate most of the dissimilarities (Figure 2). These variations in the genomic sequences seem to render characteristic DNA regulatory elements within TsA and TsB, which may justify the different transcriptional activity of these telomeres, and a more fine-tuned regulation of the expression of the TsA [24]. Additionally, these differences in sequences are enough to critically modify the structural folding of the corresponding telomeric ncRNAs.

Preliminary secondary structure predictions established the existence of specific well-defined structural regions in CriTER-A and CriTER-B that are consistent with the experimental SHAPE results. Further, conformational analysis revealed that two conformers (A1/A2 and B1/B2) could be detected for both telomeric transcripts, one of them (A1 and B1) being much more prevalent (~80%). Their relative abundance does not depend on the RNA concentration or the presence of Mg^2+^. Significantly, the proportion of the A2 conformer notably increases under thermal stress conditions (35 °C) in a Mg^2+^-dependent way whereas no differences in the CriTER-B conformers were detected. Thus, CriTER-B shows high conformational homogeneity with a compact and likely static structure and a well-defined architecture with three major stem-loops separated by single-stranded regions (Figure 7) in contrast to CriTER-A, which displays an interesting temperature-induced conformational variability. The defined domain 2 in CriTER-A, delimited for a well-defined double-stranded core (Figure 7), seems to be a relevant temperature-sensing element in this molecule.

Taken together, these results suggest that the structural tuning observed for the CriTER-A variants could be a key piece in the molecular mechanism underlying the CriTER-A function as a thermal sensor and point to A2 as the main functionally important conformer under heat stress conditions. As far as we know, this is the first time that distinguished structural features correlated to temperature changes of two homologue telomeric transcripts have been reported. Thus, these observations are an important step towards enhancing our understanding of telomeric ncRNA function in relation to cellular stress response.

## 4. Materials and Methods

### 4.1. Sequences and Plasmids

The TsA and TsB sequences are available in GenBank, Accession nos. M33211 and AJ295633, respectively. The sequences for the sense transcripts CriTER-A and CriTER-B correspond to the reverse-complementary ones of these both telomeric sequences. The plasmids with one monomeric unit of each transcript were ordered from GenScript (Netherlands). The receipt vector was pUC57. *Eco*RI sites were added in both extremes to facilitate later work.

### 4.2. DNA Templates and RNA Synthesis

DNA templates encompassing the sequence coding for CriTER-A and CriTER-B transcripts were obtained by PCR from their respective plasmids with the oligonucleotides (see Table 1) T7pCriTERA and asCriTERAB to generate the construct T7pCriTER-A and T7pCriTERB and asCriTERAB to yield the construct T7pCriTER-B. The resulting RNA molecules bore the correct 3′ ends.

For SHAPE assays and 2′-hydroxyl molecular interference (HMX) analyses, DNA templates coding for the transcripts CriTER-A+cas and CriTER-B+cas were also obtained by amplification. In this case, the RNA transcript design required slight modifications in order to optimise the recovery of the SHAPE reactivity data. Hence, two different cassettes at the 5′ and the 3′ ends were included, flanking the central sequence encoding CriTER-A or CriTER-B. These cassettes do not interfere with the structural dynamic of the transcripts (data not shown). The cassette located at the 5′ end is 48 nts long and shows an autonomous folding. It allows for the precise monitoring of SHAPE reactivities at the 5′ end of the RNA molecule under study since it moves away from the fluorescent peak corresponding to the full-length product. At the 3′ end, the 45 nts 3′ cassette provides an annealing site for efficient primer extension. Briefly, molecules T7pCriTER-A+cas and T7pCriTER-B+cas bearing these cassettes were obtained by amplification from their corresponding plasmids. Oligonucleotides T7p_CriTERA_5cas and asCriTERAB_3cas generated the T7pCriTER-A+cas template, while oligonucleotides T7p_CriTERB_5cas and asCriTERAB_3cas allowed for the production of the T7pCriTER-Bcas template (see Table 1).

RNA synthesis was accomplished with the generated DNA templates using the TranscriptAid T7 High Yield Transcription Kit (ThermoFisher Scientific), following the manufacturer’s instructions. The resulting RNA products were purified, and their quality was tested as previously described [31].

### 4.3. Conformational Analysis by EMSA

The analysis of structural heterogeneity for CriTER-A and CriTER-B monomers was performed in a dose-dependent manner. Briefly, increasing concentrations (1–100 nM) of the ^32^P-labeled RNA were denatured for 2 min at 95 °C and subsequently transferred to ice for 15 min. RNA molecules were then incubated at 20 or 35 °C in folding buffer (HEPES 100 mM, pH 8, NaCl 100 mM), either in the presence or absence of 1 mM MgCl_2_, for 30 min. RNA isoforms were immediately resolved in 6% nondenaturing polyacrylamide gels (acrylamide:bisacrylamide 19:1). Electrophoresis was performed at 4 °C and 12 V/cm, either in TBM buffer (45 mM TrisHCl at pH 8.3, 43 mM boric acid, and 0.1 mM MgCl_2_) for analyses performed in the presence of Mg^2+^; or in TBE buffer (100 mM Tris-HCl, pH 7.6, 45 mM boric acid, 2.5 mM EDTA) for those studies carried out in the absence of Mg^2+^. Gels were dried, scanned, and analysed as previously described [32].

Competition assays were performed with 10 nM of the labelled CriTER-A or CriTER-B constructs in the presence of a molar excess of the corresponding unlabelled transcripts (50, 100, 500, and 1000 nM). Briefly, RNA molecules were denatured and renatured as noted above and mixed at the indicated concentrations in the presence of folding buffer. Reactions proceeded at 20 °C for 30 min. Complexes were resolved in TBE buffer as described above. Gels were dried and scanned as reported in [32].

### 4.4. SHAPE Analysis

SHAPE analysis of CriTER-A and CriTER-B folding was performed essentially as previously described [29,33]. For this purpose, a self-folding cassette consisting of a 28 nts RNA sequence, which theoretically does not interfere with the folding of the RNA constructs, was placed at the 3′ end of each RNA.

For CriTER-A, 10 pmol of purified RNA were denatured for 2 min at 95 °C and then cooled on ice for 15 min. The RNA was then incubated at 35 °C in folding buffer with 1 mM MgCl_2_ for 5 min. Probing reactions were initiated by the addition of 15 mM of freshly diluted NMIA in DMSO. Chemical probing proceeded for 30 min at 35 °C and was stopped by snap cooling and ethanol precipitation. Control reactions were carried out in parallel in the absence of NMIA, with an equivalent volume of DMSO. The RNA was washed once with 80% ethanol, and different CriTER-A conformers were resolved by nondenaturing polyacrylamide gels as described above. Specific conformers were visualised by UV shadowing, eluted, and purified as previously described [34]. One pmol of RNA was then subjected to primer extension to detect the modified residues with 20 U of SuperScript III RT (Invitrogen, Life Technologies) in the presence of 2 pmol of the primer asHCV-372 (Table 1). This oligonucleotide, which anneals within the cassette placed at the 3′ end of transcripts, was fluorescently labelled with NED (to read treated and untreated samples), VIC, or FAM (for sequencing reactions). A fraction of the resulting cDNA samples was purified and resolved as previously reported [35]. Electropherograms were analysed using the QuShape software [36], which rendered the normalised SHAPE reactivity values for each position.

Shape-dif methodology for the CriTER-B variant was carried out as reported [29,37]. Five pmol of the RNA transcript were denatured and refolded in folding buffer as described above. Then, RNA was probed with 15 mM of NMIA or 20 mM of 1M6 for 30 and 5 min, respectively, at 35 °C. Control samples were prepared in parallel with an equivalent volume of DMSO. Reactions were stopped on ice, and the RNA was precipitated with ethanol, washed twice with 80% ethanol, and subjected to primer extension as noted above. Normalised reactivity values at each position obtained for 1M6-treated samples were then scaled up to those obtained for NMIA treatments by minimising the reactivity difference over a 25 nt sliding window and then subtracted, as reported [29], to create the SHAPE-dif profiles. For this purpose, the differential SHAPE tool was employed [38].

### 4.5. Secondary Structure Prediction

There are different implementations, which enable us to estimate the RNA secondary structure from a single sequence. The most popular are Mfold/Unafold [39] (http://www.unafold.org/, accessed on: 25 August 2021), RNAFold, included in Vienna RNA package [40] (http://rna.tbi.univie.ac.at/, accessed on: 25 August 2021), and RNAStructure [41] (http://rna.urmc.rochester.edu/, accessed on: 25 August 2021). Preliminary RNA secondary structure predictions for this study were performed using RNAStructure web server (https://rna.urmc.rochester.edu/RNAstructureWeb/, accessed on: 25 August 2021) [41,42]. MFE and a set of suboptimal structures, maximum expected accuracy structure, and pseudoknot predictions were performed with version 6.0.1 using default parameters. VARNA applet was used to edit and draw MFE structures [43].

The prediction of the theoretical RNA secondary structure models taking the SHAPE mapping date was generated using ShapeKnots [30], including the structural constraints derived from the NMIA relative reactivity data.

## Figures and Tables

**Figure 1 ijms-22-10310-f001:**
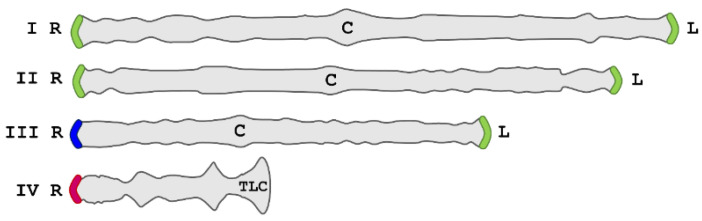
*Chironomus riparius* genome. Schematic representation of the *Chironomus riparius* chromosomes (I–IV). All telomeres except the telocentric end of the chromosome 4 (TLC) consist of large blocks of 176-bp repeat sequences. Three subfamilies of telomeric sequences have been described, which are differently distributed: TsA (blue) in telomeres III-R, TsC (pink) in telomeres IV-R, and TsB (green) in the other nontelocentric telomeres. Telomeric ncRNAs studied in this work, CriTER-A and CriTER-B, are the sense TsA and TsB transcripts, respectively. No TsC or TLC transcripts have been detected [24]. R, right; L, left; C, centromere.

**Figure 2 ijms-22-10310-f002:**
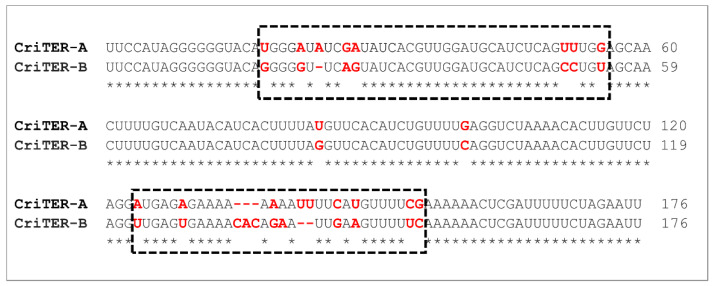
Sequence alignment of monomeric units of the telomeric CriTER-A and CriTER-B ncRNAs. Pairwise sequence alignments performed with EMBOSS Water Web services (default options, https://www.ebi.ac.uk/Tools/psa/emboss_water/, accessed on: 25 August 2021). Sequence variations are denoted in red. Two regions accumulating most of the dissimilarities are highlighted with a dashed line box.

**Figure 3 ijms-22-10310-f003:**
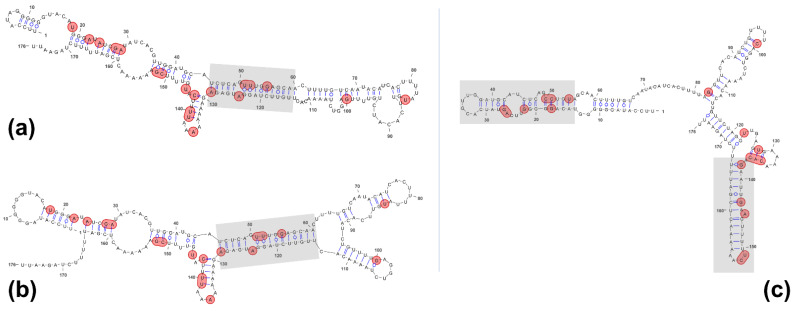
Minimum free energy (MFE) structures predicted with RNAStructure 6.0.1. (**a**) CriTER-A MFE at 20 °C; (**b**) CriTER-A MFE at 35 °C; (**c**) CriTER-B MFE. No significant temperature-dependent changes are found in MFE structures predicted for CriTER-B. Sequence variations are denoted in red. Regions with high base pairing probabilities and highly prevalent in the suboptimal spaces are shadowed (see Appendix A).

**Figure 4 ijms-22-10310-f004:**
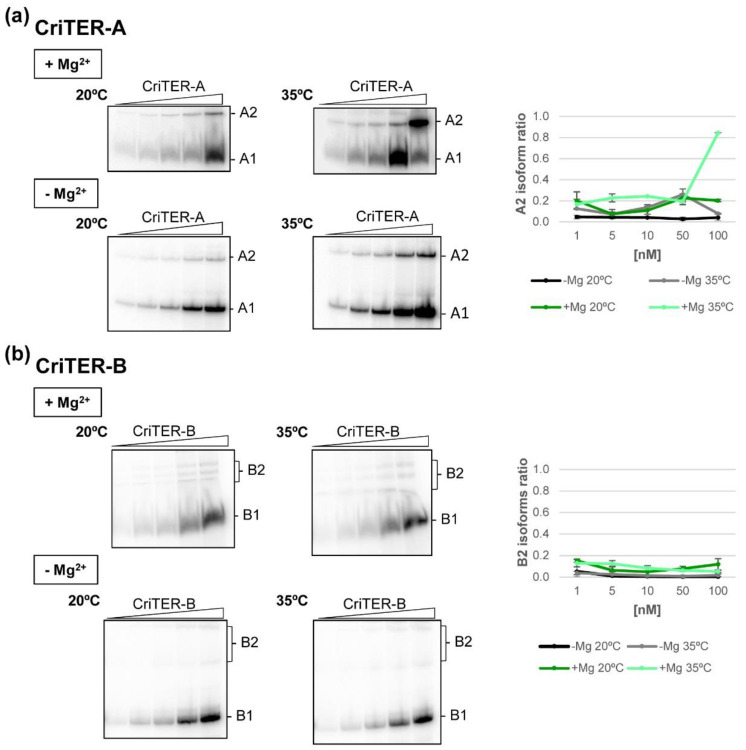
Conformational heterogeneity of CriTER-A and CriTER-B telomeric RNAs. Autoradiography of representative native polyacrylamide gels resolving increasing concentrations of CriTER-A (**a**) and CriTER-B (**b**) RNAs at basal temperature (20 °C) and under heat sock conditions (35 °C) in the presence (+) and absence (−) of 1 mM Mg^2+^. A quantification of the representative gels is shown at the right of the figure. A1 and A2 denote the two conformers of the CriTER-A RNA. B1 and B2 denote the two conformers of the CriTER-B RNA. Data represent the mean of three independent experiments.

**Figure 5 ijms-22-10310-f005:**
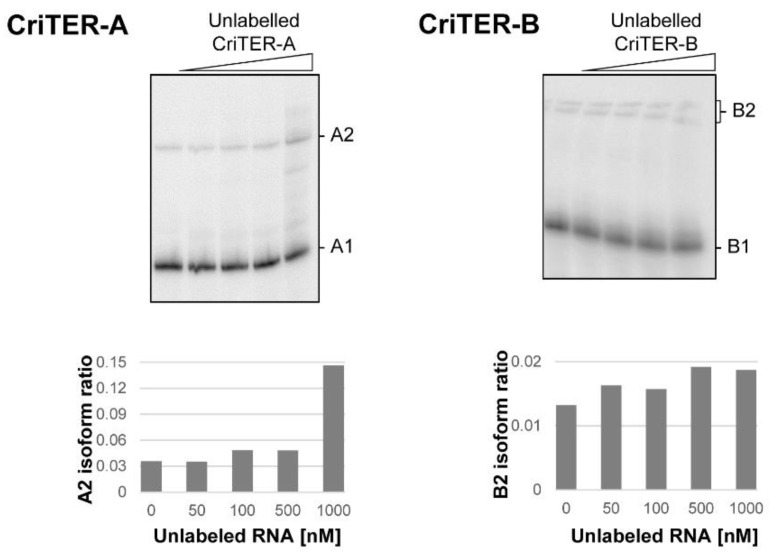
CriTER-A and CriTER-B telomeric RNAs do not dimerize. Autoradiography of native polyacrylamide gels of ^32^P-internally labelled CriTER-A and CriTER-B transcripts in the presence of increasing concentrations of the respective unlabelled RNAs. Assays were performed at basal temperature (20 °C) in the absence of Mg^2+^. A quantification of the representative gels is shown at the bottom of each panel. A1 and A2 indicate the two conformers of the CriTER-A RNA. B1 and B2 denote the two conformers of the CriTER-B RNA.

**Figure 6 ijms-22-10310-f006:**
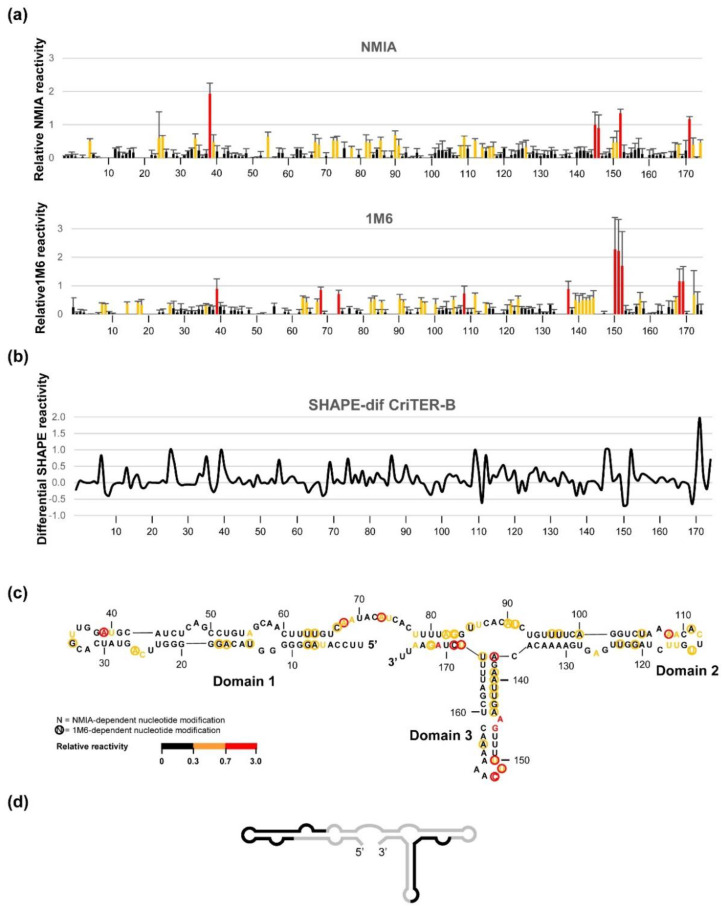
Structural analysis of the CriTER-B telomeric transcript by differential SHAPE. (**a**) Histograms show normalised reactivity of the CriTER-B monomeric RNA with NMIA (top) and 1M6 (bottom) in the presence of Mg^2+^ ions at 35 °C. Colours denote the relative nucleotide reactivity from black (lowest) to red (highest) as indicated at the bottom of the figure. Values are the mean of at least three independent experiments ± standard deviation. (**b**) Differential SHAPE reactivities (SHAPE-dif) calculated as described in [29]. (**c**) Putative structure for CriTER-B predicted by RNAStructure based on SHAPE data. (**d**) Schematic secondary structure of CriTER-B. Solid black lines represent regions containing the variable sequence stretches indicated in Figure 2.

**Figure 7 ijms-22-10310-f007:**
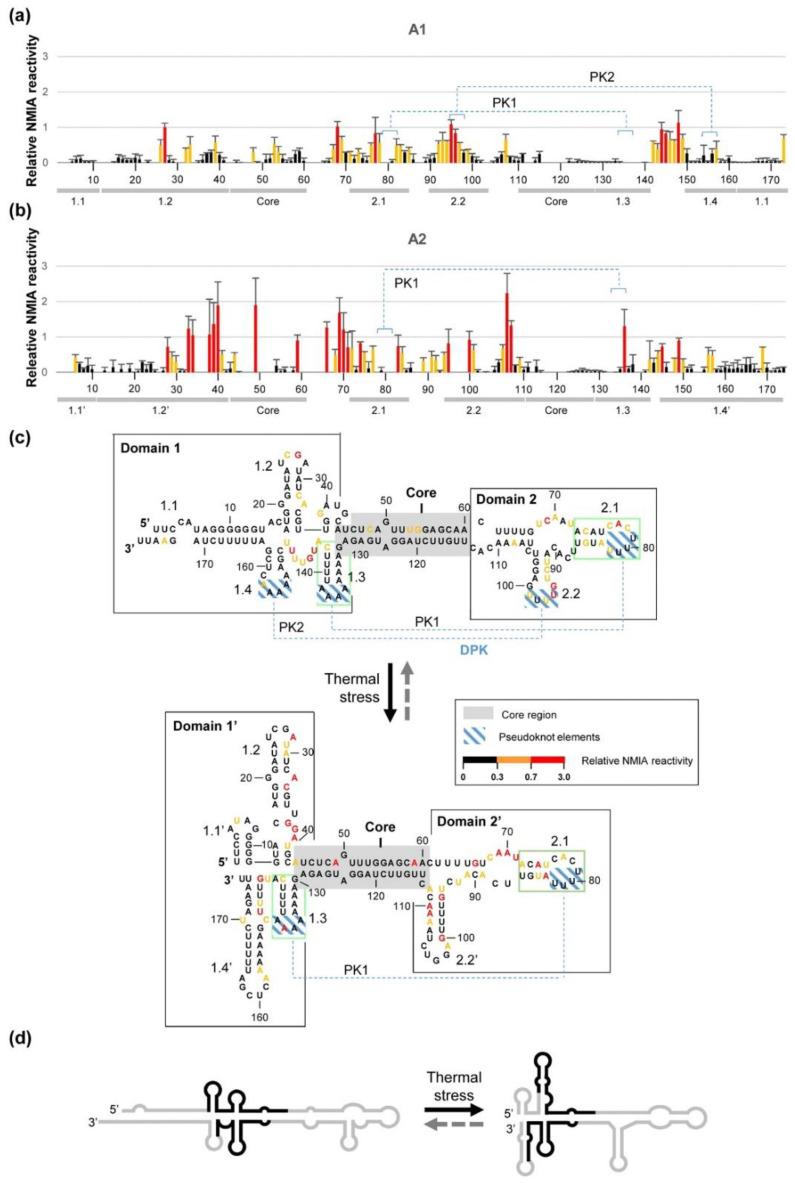
Structural analysis of the CriTER-A telomeric transcript by SHAPE. SHAPE reactivity of isolated conformers A1 (**a**) and A2 (**b**) of the CriTER-A telomeric transcript in the presence of Mg^2+^ ions at 35 °C. Histograms show normalised reactivity. Colours denote relative nucleotide reactivity from black (lowest) to red (highest). Values are the mean of at least three independent experiments ± standard deviation. (**c**) Putative structure for CriTER-A conformers (A1 top, A2 bottom) predicted by RNAStructure based on SHAPE data. The core region is boxed in grey. Green boxes delimit those structurally preserved areas in both conformers. (**d**) Schematic representation of the secondary structure of CriTER-A, indicating with solid black lines the variable sequence stretches detected by comparative sequence alignment as described in Figure 2.

**Table 1 ijms-22-10310-t001:** Oligonucleotides used in this study.

**Title 1** **Oligonucleotide**	**Sequence (5′-3′) ^1^**
T7pCriTERA	TAATACGACTCACTATAGGGTTCCATAGGGGGGTACATGGGATA
T7pCriTERB	TAATACGACTCACTATAGGGTTCCATAGGGGGGTACAGGGGGTT
asCriTERAB	AATTCTAGAAAAATCGAGTTT
T7p_CriTERA_5cas	TAATACGACTCACTATAGGGACCAACCGGCGCGCCCACAGGACGTCAA GTTCCCGGGCCGTGGTCAGATTCCATAGGGGGGTACATGGGATA
T7p_CriTERB_5cas	TAATACGACTCACTATAGGGACCAACCGGCGCGCCCACAGGACGTCAA GTTCCCGGGCCGTGGTCAGATTCCATAGGGGGGTACAGGGGGTT
asCriTERAB_3cas	TTTTTCTTTGAGGTTTAGGATTCGTGCCAGTGGTGCACGGTCTACAATTCT AGAAAAATCGAGTTT

^1^ T7 promoter is underlined. 5′ and 3′ cassettes included in the SHAPE and HMX analyses are denoted by red and blue lettering, respectively.

## Data Availability

All the data are included in the figures and in the Appendix A.

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
