# Peer review of "CriTER-A: A Novel Temperature-Dependent Noncoding RNA Switch in the Telomeric Transcriptome of Chironomus riparius"

_ijms, 2021, doi:10.3390/ijms221910310_

Round 1
Reviewer 1 Report
In this manuscript, Romero-Lopez and colleagues investigate the structure of two homologous telomeric transcripts expressed in C. riparius: CriTER-A and CriTER-B. Previous evidence indicate that CriTER-A is induced upon heat shock, while CriTER-B is only slightly affected by thermal stress. The authors hypothesized that structural differences between CriTER-A and CriTER-B could result in different functionalities during thermal stress.
The authors perform interesting computational approaches to provide evidence of potential distinctive folding of the CriTER-A and CriTER-B ncRNAs upon heat-shock. These analyses predicted a conformational change of CriTER-A at 35°C which was supported by SHAPE analysis and an in vitro EMSA assay showing a shift in mobility of a CriTER-A mimicking oligonucleotide when incubated at 35° at high concentration. However, the authors fall short in confirming their results by in vivo experimental approaches and it is very possible that the conformations predicted and observed in vitro will not occur in vivo.
Main concerns:
SHAPE is an interesting approach very appropriate to assess RNA conformation. However, the authors analyse only in vitro transcribed RNAs. The authors should try to apply this technique to RNA extracted from cells or organisms, as recently done by Liu et al., Genome Biology 2021 in Arabidopsis (https://doi.org/10.1186/s13059-020-02236-4). The authors could try to pull down endogenous CriTER-A and CriTER-B RNA using biotinylated antisense oligonucleotide and then assess them by SHAPE.
C. riparius telomeric RNAs can contain a monomer or multimers of the telomeric sequence. All the analyses done by the authors, in silico, SHAPE and EMSA, were performed considering or using CriTER-mimicking oligo ribonucleotides of a specific length, corresponding to the RNAs comprising a monomer of the telomeric sequence. We do not know whether they recapitulate the behavior of the transcripts in vivo. Furthermore, monomers represent the minority of the telomeric transcripts, what about the rest of the telomeric molecules? how do they fold in response to heat shock? If monomers of telomeric transcripts are only a minor fraction of the total telomeric transcripts in the cells, it is difficult to ascribe any biological effect to such isoform shift under heat shock conditions. The authors could at least perform the same experiments using RNAs containing two telomeric sequences. Again, the approaches should be attempted also on cellular RNA.
Figure 4. RNAs incubated at 20 °C and 35 °C should be run on the same gel to better appreciate the differences in mobility upon heat shock. In Figure 4A, it is unclear why the shift between the two isoforms of CriTER-A occurs only at the highest concentration. Are these concentrations recapitulating the physiological concentration of the same RNA in the cell under heat shock conditions?
Introduction section:
Line 90-93: “As far as we know, Chironomus riparius is the sole biological system in which differential transcriptional activity in the telomeres has been detected under different environmental conditions rendering two homologous ncRNAs, CriTER-A and CriTER-B”. I would be cautious with such statements because the subtelomeric regions of TERRA are homologous to one another and TERRA transcripts expressed from different telomeres are well known to be regulated by different mechanisms, also in response to environmental conditions; Furthermore, in addition to TERRA mammalian telomeres express tdlincRNAs which can be both sense and antisense to TERRA and they are regulated by alternative mechanisms.
Lines 82-83: Even if specific transcription factor binding sites have been identified in TsA and TsB sequences, yet unidentified promoter sequences in subtelomeric regions could participate to the distinct response of the telomeric transcripts to environmental stress.
It would be important to give some background on the telomere length homeostasis of C. riparius
Author Response
Reviewer 1
“SHAPE is an interesting approach very appropriate to assess RNA conformation. However, the authors analyse only in vitro transcribed RNAs. The authors should try to apply this technique to RNA extracted from cells or organisms, as recently done by Liu et al., Genome Biology 2021 in Arabidopsis (https://doi.org/10.1186/s13059-020-02236-4). The authors could try to pull down endogenous CriTER-A and CriTER-B RNA using biotinylated antisense oligonucleotide and then assess them by SHAPE.”
We agree with the reviewer in the relevance of exporting this in vitro data to in vivo. Besides on the interesting strategy proposed by the reviewer, we have accumulated some experience on SHAPE in vivo by directly treating cells with SHAPE reagents. After total RNA extraction, RNA modifications can be specifically detected by primer extension and further RNA-Seq. This approach was proposed with modifications by two different groups, Weeks’ lab and Lucks’ lab (Watters et al., 2016, Methods, 103:34-48; Smola and Weeks, 2018. Nature Prot. 13:1181-1195). These strategies have been extensively used, which confirms their usefulness. However, as the reviewer can realize, this protocol needs extensive implementation, which takes several months. Therefore, we consider that it is not affordable to include in the current manuscript. We have tried to make it clear throughout the manuscript that this is an in vitro study.
We really appreciate the comment of the reviewer and we will keep it in mind for subsequent studies.
“C. riparius telomeric RNAs can contain a monomer or multimers of the telomeric sequence. All the analyses done by the authors, in silico, SHAPE and EMSA, were performed considering or using CriTER-mimicking oligo ribonucleotides of a specific length, corresponding to the RNAs comprising a monomer of the telomeric sequence. We do not know whether they recapitulate the behavior of the transcripts in vivo […] The authors could at least perform the same experiments using RNAs containing two telomeric sequences.”
It is an excellent observation. We concur with the referee that we cannot know whether monomers will reflect the folding of concatemers in vivo. However, the proposed folding seems to be quite autonomous and therefore it could be extrapolated to CriTER polymers. On the other hand, it is also likely that only monomeric molecules can perform regulatory functions. In this case, the preservation of the folding in CriTER polymers would not be a requisite.
“Furthermore, monomers represent the minority of the telomeric transcripts, what about the rest of the telomeric molecules? How do they fold in response to heat shock? If monomers of telomeric transcripts are only a minor fraction of the total telomeric transcripts in the cells, it is difficult to ascribe any biological effect to such isoform shift under heat shock conditions.“
We thank the reviewer for these interesting observations. Nevertheless, it must be noted that the function performed by RNA molecules is fine regulated and depends on complex conformational equilibriums in which transitions between different structures may displace the balance toward one side or another. It is assumed that high concentrations of regulatory RNAs are not necessarily linked to an efficient response, since RNA regulatory function can indeed be performed by different mechanisms such as RNA-protein interactions, intermolecular RNA-RNA interactions or RNA structural rearrangements. Further, different molecular mechanisms can underlie the same function accomplished by an RNA molecule. The existence of different molecular mechanisms provides a reliable biological support since it allows for taking advantage of specific kinetic and thermodynamic behaviours of the RNA molecules, thus reducing the need of huge amount of the regulatory compound. This is the base of the stochasticity principle, which is an important property of biological regulatory systems (Vandevenne et al, 2019, Front. Genet. Vol. 10: article 403, and references therein).
“Figure 4. RNAs incubated at 20 °C and 35 °C should be run on the same gel to better appreciate the differences in mobility upon heat shock.”
We concur with the reviewer in this point, but gels were loaded with the aim of comparing mobility between CriTER-A and CriTER-B, since this can inform about differences in conformations between both variants. Thus, we can obtain structural information able to support those variations in the folding obtained from the SHAPE analysis.
“In Figure 4A, it is unclear why the shift between the two isoforms of CriTER-A occurs only at the highest concentration. Are these concentrations recapitulating the physiological concentration of the same RNA in the cell under heat shock conditions?”
This phenomenon suggests that the conformational rearrangements in CriTER-A are concentration-dependent. Unfortunately, these assays do not recapitulate the in vivo conditions. It seems likely that the presence of protein factors like chaperones, ions or cofactors may influence this equilibrium, favouring the acquisition of the A2 conformation at lower RNA concentrations than those reported in vitro. Further assays will be required to identify these adjuvants.
“Line 90-93: “As far as we know, Chironomus riparius is the sole biological system in which differential transcriptional activity in the telomeres has been detected under different environmental conditions rendering two homologous ncRNAs, CriTER-A and CriTER-B”. I would be cautious with such statements because the subtelomeric regions of TERRA are homologous to one another and TERRA transcripts expressed from different telomeres are well known to be regulated by different mechanisms, also in response to environmental conditions; Furthermore, in addition to TERRA mammalian telomeres express tdlincRNAs which can be both sense and antisense to TERRA and they are regulated by alternative mechanisms.”
Thank you for these comments. We agree, differential transcriptional activity in response to environmental conditions has been well established. In this sentence, we wanted to point out the existence of two reported homologues ncRNAs with substantial analysed differences in their sequences. We consider interesting to highlight the interest of study the putative variation in the sequence for the different homologue telomeric/subtelomeric RNAs, to characterise each of these molecules at sequence level, and to study their transcriptional activity. These sentences have been rewritten.
“Lines 82-83: Even if specific transcription factor binding sites have been identified in TsA and TsB sequences, yet unidentified promoter sequences in subtelomeric regions could participate to the distinct response of the telomeric transcripts to environmental stress.”
We concur with the reviewer that it could be a possibility, which could be evaluated in future studies. Nevertheless, transcriptional activity of subtelomeric regions in these insects has not been reported, and HSF and RNA pol II HSF localize in telomeric puff during heat shock, suggesting that the promoter is in the telomere and not in the sub-telomeric region (Morcillo et al. 1994, Martinez-Guitarte et al. 2008). On the other hand, the transcriptional activity of the telomeric region has been deeply studied, and the internal promoter in telomeric region seems clear, either from in silico studies, or from experimental assays, which showed the binding of Heat Shock Factors in these regions (Morcillo et al., 1994, Exp Cell Res. 211:163-7; Martínez-Guitarte et al., 2008, Chromosome Res. 16:1085–1096).
“It would be important to give some background on the telomere length homeostasis of C. riparius.”
It is complex to provide the background about the telomere length. Most of the studies are done in salivary gland cells and with fourth instar larvae. In the first case, it is a differentiated tissue while in the second it is an advanced stage. There are few data about the rest of developmental stages, which can provide some insights in the dynamic of the telomere. Right now, it is only known that a putative retrotranscriptase could be involved (detected by in situ immunofluorescence) but there is no real clue about the mechanisms of maintaining the telomere length and how it is modulated along the time in the different development stages.

Reviewer 2 Report
Manuscript ID: ijms-1374470
Title: CriTER-A: a novel temperature-dependent noncoding RNA switch in the telomeric transcriptome of Chironomus riparius
The manuscript describes an analysis of thermal-stress driven structural changes in two telomeric transcripts from Chironomus riparius. The authors present a comprehensive (and well written) introduction to the topic of non-coding RNA telomeric transcripts. Further, they show results supporting an hypothesis that CriTER-A, one of the analysed ncRNAs, is an RNA thermometer of yet-unknown regulatory function. I think that the manuscript can be recommended for publication in the International Journal of Molecular Sciences after authors address several issues listed below.
The very first results presented in the manuscript are predicted structures of the two (tentative) regulatory elements CriTER-A and CriTER-B. This clearly suggests that the former may undergo conformational changes at elevated temperatures, which is further confirmed with electrophoretic mobility shift assays results. However, a later, and more reliable analysis based on experimental probing (SHAPE) suggests that the actual secondary structure of the two transcripts differ from the theoretical predictions. I think it is very confusing for a reader and the authors should either remove theoretical secondary structure predictions from the manuscript (as less reliable) or discuss in detail differences between the models.
Moreover, I was very surprised that competition assays showing that the slow migrating transcripts are not dimeric were not presented (one of the most important results in this work!). These results should be added to the manuscript (or at least to supplementary materials) and discussed in more detail.
I also wonder, why the final analysis of the two CriTER-A conformers was performed at 35 degrees Celsius only (after fractionation). Does the SHAPE analysis at 20 degrees agree with the A1 confirmation revealed at high temperature? Was such an analysis done?
Finally, it is not entirely clear to me why the predicted differences in CriTER-A secondary structure result in such dramatic changes in mobility shift essays. I think that the authors should comment on this. I also wonder if it wouldn’t be helpful here to predict 3D structures from SHAPE-assisted secondary structures using any of the available computational methods.
Author Response
Reviewer 2
“The very first results presented in the manuscript are predicted structures of the two (tentative) regulatory elements CriTER-A and CriTER-B. This clearly suggests that the former may undergo conformational changes at elevated temperatures, which is further confirmed with electrophoretic mobility shift assays results. However, a later, and more reliable analysis based on experimental probing (SHAPE) suggests that the actual secondary structure of the two transcripts differ from the theoretical predictions. I think it is very confusing for a reader and the authors should either remove theoretical secondary structure predictions from the manuscript (as less reliable) or discuss in detail differences between the models.”
As the reviewer points, structures solved based on experimental probing SHAPE differ from theoretical predictions. This was certainly what it was expected. Current parameters used in theoretical predictions are expected to predict about the 70% of base pairs correctly, on average, although it varies widely. On the other hand, base-pairs probabilities and suboptimal structures studies can help to identify “well-defined” regions, which are usually predicted with much higher accuracy. Our predictions show some regions exhibiting high base-pair probabilities and highly prevalent in the suboptimal spaces (see Figure 3 and supplementary material). The predicted base pairs in these regions agree with the results of SHAPE experiment. Thus, we think that it is worth to include these theoretical results, since they were key preliminary information in the study. Nevertheless, we concur that the most relevant part are the experimental results. This is why we include most of the theoretical results in the supplementary material, despite they were very relevant to think and design the experimental part of our work.
“Moreover, I was very surprised that competition assays showing that the slow migrating transcripts are not dimeric were not presented (one of the most important results in this work!). These results should be added to the manuscript (or at least to supplementary materials) and discussed in more detail.”
We appreciate this suggestion. Further data have been added to the main text and discussed accordingly.
“I also wonder, why the final analysis of the two CriTER-A conformers was performed at 35 degrees Celsius only (after fractionation). Does the SHAPE analysis at 20 degrees agree with the A1 confirmation revealed at high temperature? Was such an analysis done?”
We isolated both conformations, A1 and A2, which can be found at 35ºC, to perform the SHAPE assays. Since the mobility of A1 was not affected at 20 and 35ºC, we assumed that the structure is the same at both temperature conditions.
“Finally, it is not entirely clear to me why the predicted differences in CriTER-A secondary structure result in such dramatic changes in mobility shift essays. I think that the authors should comment on this.”
We agree with the reviewer that changes in mobility are impressive. It must be noted that electrophoretic mobility shift assays are a well-established method to study conformational kinetics and folding equilibrium constants. Importantly, under non-denaturing conditions, it is impossible to predict how a conformational rearrangement can influence the mobility of the molecule. It has been largely documented that depending on salt concentrations, acrylamide:polyacrylamide ratio, …, the mobility of an RNA molecule can be dramatically affected (Woodson and Koculi, 2009, Methods Enzymol., 469: 189–208). In addition, it has been demonstrated that tightly packed RNA molecules migrate faster than unfolded or relaxed RNA conformations (Woodson and Koculi, 2009, Methods Enzymol., 469: 189–208). We consider that these observations perfectly fit with the obtained data. We have added a comment in the manuscript to clarify this.
“I also wonder if it wouldn’t be helpful here to predict 3D structures from SHAPE-assisted secondary structures using any of the available computational methods.”
This is a very good point. We absolutely agree with the reviewer, as he/she suggests the global structure of the RNA molecule determines its mobility. Unfortunately, 3D structural modelling is not simple and the available softwares do not provide as accurate results as expected from SHAPE reactivity data.

Reviewer 3 Report
In this manuscript, authors analyzed the impact of heat shock conditions on the folding of two telomeric transcripts from Chironomus riparius, named CriTER-A and CriTER-B. Despite a significant homology between the two RNA sequences, authors identified consistent structural differences between CriTER-A and CriTER-B. Moreover, they observed a structural switch of CriTER-A under heat shock conditions that could envisage some biological implication. The herein described results could rise the interest in the field of ncRNAs and telomere regulation, however the manuscript requires some revision before being accepted for publication.
- The major point concerns Figure 4a, where authors reported the EMSA assays of CriTER-A RNA in normal and heat shock conditions. In the presence of magnesium ions at 35°C, species A1 and A2 switch their relative abundance and this phenomenon is due to the increase of CriTER-A concentration. Thus, A2 folding seems to be concentration-dependent, but authors exclude the formation of dimers based on competition assays that are not shown. I think it is necessary including these data to clarify the findings for the readers.
- EMSA assay were performed (or reported at least) without any molecular weight marker. It would be good to include a reference. Moreover, it would be good if the authors could add the full gels in the supplementary materials.
- The gel quantification reported in Figure 4a-b is very small and difficult to read. I think it should be improved.
- Preliminary structural models for CriTER-A and -B were predicted by applying RNAStructure 6.0.1. Is this tool considering the formation of non-canonical structures in the analysis?
- English language and style require some editing.
Author Response
Reviewer 3
“The major point concerns Figure 4a, where authors reported the EMSA assays of CriTER-A RNA in normal and heat shock conditions. In the presence of magnesium ions at 35°C, species A1 and A2 switch their relative abundance and this phenomenon is due to the increase of CriTER-A concentration. Thus, A2 folding seems to be concentration-dependent, but authors exclude the formation of dimers based on competition assays that are not shown. I think it is necessary including these data to clarify the findings for the readers.”
We thank the reviewer for this suggestion. We have added new data illustrating this point.
“EMSA assay were performed (or reported at least) without any molecular weight marker. It would be good to include a reference.”
Since EMSA are performed under non-denaturing conditions, RNA mobility mainly depends on its folding. Therefore, adding a molecular weight marker is not a good option since its migration will not depend on the length of the RNA products, but it will depend on their conformation. Thus, the results could be misunderstood.
“Moreover, it would be good if the authors could add the full gels in the supplementary materials.”
Full gels have been added to the submission platform.
“The gel quantification reported in Figure 4a-b is very small and difficult to read. I think it should be improved.”
We thank the reviewer for this suggestion. We have modified the figure accordingly.
“Preliminary structural models for CriTER-A and -B were predicted by applying RNAStructure 6.0.1. Is this tool considering the formation of non-canonical structures in the analysis?”
In this work, we have use RNAstructure Web Server, which combine “partition”, “Fold”, “MaxExpect” and “ProbKnot” algorithms, which, as far as we know, consider only canonical base pairs. On the other hand, the RNAStructure software package provide a method (CycleFold), that can identify non-canonical base pairs, but we have not applied it in this study. Considering the formation of non-canonical interactions will be relevant and interesting in future studies, especially for accomplishing the prediction of the full tertiary structure.
“English language and style require some editing.”
English language has been revised by a professional proofreading and editing service. The corresponding certificate is included.

Reviewer 4 Report
The manuscript by Romero-Lopez et al. presents a novel temperature-dependent noncoding RNA switch in the telomeric transcriptome of Chironomus riparius. There are two telomeric transcripts, so called CriTER-A and CriTER-B variants, both transcripts showing high sequence homology and are studied for their folding and its influence on their differential behavior. The study suggests that CriTER-A folds as two different conformers, suggesting a putative functional role of the telomeric transcriptome in relation to thermal stress that could rely on CriTER-A.
The manuscript is interesting and combines experimental and computational methods. While I am not an expert on telomeric transcirptome and my expertise is on the bioinformatics side, it appears to me novel and with biological significance. My comments are geared towards improving the presentation of the manuscript from the bioinformatics perspective. The major comments I regard as essential.
Major Comments:
1) There are limitations when using the temperature knob in RNA folding prediction. As written in the RNAfold webserver (I recommend the authors look it up in the Vienna RNA package) in the help page for temperature, "The energy parameters used in the calculation have been measured at 37C. Parameters at other temperatures can be extrapolated, but for temperatures far from 37C results will be increasingly unreliable". The authors at the very least need to mention this limitation along with a brief discussion.
2) The authors rely in their manuscript on RNA folding prediction by energy minimization. It is fine to use the RNAStructure web server for that, but the authors need to cite the Vienna RNA package and mfold as well, which were developed before RNAStructure and are at least as important in the field of RNA folding prediction. That said, RNAStructure is well respected.
3) In Figure 3, (a) and (b) and (c) are presented in a confusing manner in the caption. One needs to start with (a) and describe what it is, then (b) and describe what it is, etc. The sentence "CriTER-B MFE (c), where no temperature-dependent changes are predicted" is not clear by itself.
4) In the supplementary information, I could not find Figure S8 for the "conformational heterogeneity of CriTER-A and CriTER-B telomeric RNAs: competition assays" as displayed at the top of the file.
Minor comments:
5) In Figure S2 and S5, I recommend calling the "lineal representation" by its more common name, "arc diagram". And in Figure S6, this is a "circular representation", and this needs to be mentioned in the caption for clarity.
6) In the Introduction, for the purpose of horizon-broadening, I would recommend mentioning RNA thermometers that were discovered by Narberhaus and co-workers, perhaps RNA bioinformatics tools using folding prediction by energy minimization to discover the location of RNA thermometers such as RNAthermsw and RNAtips, possibly "RNA-mediated response to heat shock in mammalian cells" by (Shamovsky et al., 2006) in Nature where the secondary structure is available in the supplementary file, etc. I leave it for the discretion of the authors, but I think the Introduction could benefit from expanding in at least some of these directions.
7) The resolution of Figure 3 can perhaps be improved.
Author Response
Reviewer 4
“There are limitations when using the temperature knob in RNA folding prediction. As written in the RNAfold webserver (I recommend the authors look it up in the Vienna RNA package) in the help page for temperature, "The energy parameters used in the calculation have been measured at 37C. Parameters at other temperatures can be extrapolated, but for temperatures far from 37C results will be increasingly unreliable". The authors at the very least need to mention this limitation along with a brief discussion.”
That is a good point. We thank to the reviewer for highlighting it. We have remarked this limitation in the revised version of our manuscript, along with pointing out the interest of the analysis of the predicted structures for our two homologues sequences at different temperature for comparative purposed.
“The authors rely in their manuscript on RNA folding prediction by energy minimization. It is fine to use the RNAStructure web server for that, but the authors need to cite the Vienna RNA package and mfold as well, which were developed before RNAStructure and are at least as important in the field of RNA folding prediction. That said, RNAStructure is well respected.”
Thank you for this important note. We have cited Vienna RNA package and Mfold already.
“In Figure 3, (a) and (b) and (c) are presented in a confusing manner in the caption. One needs to start with (a) and describe what it is, then (b) and describe what it is, etc. The sentence "CriTER-B MFE (c), where no temperature-dependent changes are predicted" is not clear by itself.”
There was a mistake in this caption (it was mixed with caption of Figure 2). Now, it is revised and fixed.
“In the supplementary information, I could not find Figure S8 for the "conformational heterogeneity of CriTER-A and CriTER-B telomeric RNAs: competition assays" as displayed at the top of the file.”
According to suggestions of reviewers 1, 2 and 3, we have included new data in the manuscript to clarify this point.
“In Figure S2 and S5, I recommend calling the "lineal representation" by its more common name, "arc diagram". And in Figure S6, this is a "circular representation", and this needs to be mentioned in the caption for clarity.”
We have rewritten these captions according to the suggestion.
“In the Introduction, for the purpose of horizon-broadening, I would recommend mentioning RNA thermometers that were discovered by Narberhaus and co-workers, perhaps RNA bioinformatics tools using folding prediction by energy minimization to discover the location of RNA thermometers such as RNAthermsw and RNAtips, possibly "RNA-mediated response to heat shock in mammalian cells" by (Shamovsky et al., 2006) in Nature where the secondary structure is available in the supplementary file, etc. I leave it for the discretion of the authors, but I think the Introduction could benefit from expanding in at least some of these directions.”
We appreciate these suggestions. Despite we find very interesting these directions, we have not mentioned in the introduction trying not to expand too much this section. On the other hand, we have use RNAtips for subsequent work with multimers, containing two or more copies of the sequence. Preliminary results were very interesting. We will consider all these tips in this next work.
“The resolution of Figure 3 can perhaps be improved.”
We are confident that the resolution of Figure 3 will be better in final edition, by using the high-resolution image that we submitted.

Round 2
Reviewer 2 Report
In the revised version of the manuscript the authors addressed most of my comments in a way that is fully satisfactory.
Reviewer 3 Report
The manuscript got improved after the revision, but I still have some concerns about the competition assays.
Would it be possible to repeat the experiments at 35°C and in the presence of Mg2+ (the condition that authors showed as the most relevant for the formation of species A2)?